# Mesonephric Adenocarcinoma of the Vagina Harboring *TP53* Mutation

**DOI:** 10.3390/diagnostics12010119

**Published:** 2022-01-05

**Authors:** Hyunjee Lee, Hyunjin Kim, Hyun-Soo Kim

**Affiliations:** Department of Pathology and Translational Genomics, Samsung Medical Center, Sungkyunkwan University School of Medicine, Seoul 06351, Korea; hyunjee.lee@samsung.com

**Keywords:** vagina, mesonephric adenocarcinoma, *TP53* mutation, aberrant p53 expression

## Abstract

Mesonephric adenocarcinoma (MA) of the female genital tract is a rare but distinct entity, exhibiting unique morphological, immunophenotypical, and molecular characteristics. Vaginal MA is hypothesized to arise from the mesonephric remnants located in the lateral vaginal wall. A 52-year-old woman presented with vaginal bleeding. Physical examination revealed a protruding mass in the left vaginal wall. Pelvic magnetic resonance imaging revealed a 2.5-cm mass arising from the left upper vagina and extending posterolaterally to the extravaginal tissue. The punch biopsy was diagnosed as poorly differentiated adenocarcinoma. She received radical surgical resection. Histologically, the tumor displayed various architectural patterns, including compactly aggregated small tubules, solid cellular sheets, endometrioid-like glands and ducts, intraluminal micropapillae, cribriform structure, and small angulated glands accompanied by prominent desmoplastic stroma. The tubules and ducts possessed hyaline-like, densely eosinophilic intraluminal secretions. The tumor extended to the subvaginal soft tissue and had substantial perineural invasion. Immunostaining revealed positivity for the mesonephric markers, including GATA3, TTF1, and PAX2, while showing very focal and weak positivity for estrogen receptor and negativity for progesterone receptor. Additionally, we observed a complete absence of p53 immunoreactivity. Targeted sequencing analysis revealed that the tumor harbored both activating *KRAS* p.G12D mutation and truncating *TP53* p.E286* mutation. A thorough review of the previous literature revealed that 4.5% (3/67) of vaginal/cervical MAs and 0.9% (1/112) of uterine/ovarian mesonephric-like adenocarcinomas harbor *TP53* mutations, indicating that this is very uncommon in malignant mesonephric lesions. In summary, we presented a rare case of vaginal MA uniquely harboring pathogenic *TP53* mutation, resulting in p53 aberration.

## 1. Introduction

The sexually dimorphic establishment of the reproductive system is a critical step in the embryogenesis [1]. An embryo before the sexual differentiation possesses both female and male genital tract progenitors, paired Mullerian (paramesonephric) and Wolffian (mesonephric) ducts [2]. Starting about eight weeks from gestation, the embryo eliminates one of the two progenitors and maintains the other. The embryo retains only one reproductive tract corresponding to its sex: Mullerian duct for the female and Wolffian duct for the male [2]. The Mullerian ducts in the female eventually differentiate into the adult female reproductive tracts, which include the fallopian tubes, uterine corpus, cervix, and the upper part of vagina [3]. On the other hand, the Wolffian ducts in the male develop into the adult male reproductive organs, which include the epididymis, vas deferens, and the seminal vesicles. In the female, the Wolffian ducts eventually regress, but their remnants give rise to the rete ovarii, para-oophoron, and the Gartner duct. Such mesonephric remnants are usually found in the meso-ovarium, broad ligament, and the lateral wall of the cervix or vagina and may give origin to cysts and, rarely, tumors.

Mesonephric adenocarcinoma (MA) is a rare malignant tumor of the female genital tract thought to arise from the mesonephric remnants or hyperplasia [4,5]. MA typically arises in the uterine cervix and vagina, although some cases of mesonephric-like adenocarcinoma (MLA) arising in the uterine corpus and adnexa have been reported [5,6]. Since this tumor exhibits a variety of histological growth patterns and mimics more common gynecological malignancies, it can lead to incorrect diagnosis and management.

Primary vaginal carcinoma is rare, accounting for only 1–2% of all gynecological malignancies [5]. The majority of vaginal carcinomas are squamous cell carcinoma, comprising 90% of primary vaginal malignancies [7]. Approximately 5–8% of primary vaginal malignancies are adenocarcinomas [8]. Although several histological subtypes of vaginal adenocarcinomas, such as clear cell, endometrioid, serous, and mucinous adenocarcinomas, have been documented [9], MA of the vagina is exceedingly rare. Twenty-two cases of vaginal MA have been reported in the literature so far [10,11,12,13,14,15,16,17,18,19,20,21,22,23].

We recently experienced a rare case of primary vaginal MA occurring in a 52-year-old woman. While several researchers have acknowledged and investigated uterine MA and MLA, much remains unknown about the clinical and pathological features of vaginal MA. Moreover, there has not been any study examining the molecular features of vaginal MA. In this report, we aim to provide a thorough clinicopathological description of vaginal MA, as well as its immunophenotype and genetic features. In particular, we observed that this tumor harbored pathogenic tumor protein 53 (*TP53*) mutation and aberrant p53 expression, which have never been reported in previous literature on vaginal MA. For planning appropriate therapeutic strategies, it is critical for pathologists to determine the histological subtype of vaginal adenocarcinoma. Our comprehensive clinicopathological and molecular analysis can serve to improve the understanding of this rare condition and help pathologists in making an accurate diagnosis.

## 2. Case Presentation

A 52-year-old woman presented with a 3-month history of vaginal bleeding. She had no medical history. Physical examination revealed a protruding mass in the left vaginal wall. Pelvic magnetic resonance imaging revealed a 2.5-cm mass arising from the left upper vagina and extending posterolaterally to the extravaginal tissue (Figure 1). The punch biopsy was diagnosed as poorly differentiated adenocarcinoma. Under the preoperative clinical impression of vaginal carcinoma, she received a radical surgical resection of the tumor with bilateral salpingo-oophorectomy and pelvic lymph node dissection.

Grossly, the mass was a poorly circumscribed, infiltrative solid tumor with a yellow-to-white and solid-cut surface. This tumor appeared to involve the entire thickness of the vaginal wall and extend to the subvaginal soft tissue (Figure 1). Two experienced gynecological pathologists performed a detailed microscopic examination. Immunostaining and targeted sequencing were also performed to confirm the diagnosis. Histological features and immunostaining results are shown in Figure 2 and Figure 3, respectively.

Targeted next-generation sequencing (NGS) was performed using the Oncomine Comprehensive Assay v3 (Thermo Fisher Scientific, Waltham, MA, USA), a commercial, amplicon-based, targeted cancer gene panel that enables the detection of relevant single-nucleotide variants, amplifications, and indels from 161 unique genes. As shown in Table 1, NGS analysis revealed that the tumor harbored pathogenic mutations in Kirsten rat sarcoma viral oncogene homolog (*KRAS*) and *TP53*, both of which were verified in two databases, ClinVar (National Institutes of Health, Bethesda, MD, USA) and Catalogue of Somatic Mutations in Cancer (Wellcome Sanger Institute, Cambridgeshire, UK). The diagnosis of vaginal mesonephric adenocarcinoma (MA) was established by characteristic histological and immunophenotypical features, including architectural diversity, compactly aggregated small tubules, and eosinophilic intraluminal secretions; expression for multiple mesonephric markers; only focal and weak estrogen receptor and completely negative progesterone receptor expression; and, finally, activating *KRAS* p.G12D mutation. Additionally, the identification of nonsense *TP53* p.E286* mutation, leading to the formation of a truncated, non-immunoreactive protein, was consistent with the complete absence of p53 immunoreactivity.

The patient received sequential chemoradiation therapy as postoperative adjuvant treatment. Five cycles of weekly cisplatin were administered (40 mg/m^2^). She subsequently received whole-pelvic radiation therapy (5000 cGy/25 fractions), followed by intracavitary brachytherapy (1000 cGy/2 fractions). She is alive without evidence of disease at 10 months postoperatively. The serum levels of cancer antigen (CA) 125, CA 19-9, carcinoembryonic antigen, and squamous cell carcinoma antigen were within normal range. Serial chest and abdomen computed tomography revealed no evidence of recurrent tumor or metastasis. Follow-up cytology was negative for intraepithelial lesion or malignancy.

## 3. Discussion

MA is a rare malignant tumor of the female genital tract hypothesized to derive from the embryonal remnants of the mesonephric tubules and ducts [5]. MA typically arises in the lateral wall of the uterine cervix or vagina. Using the PubMed (National Library of Medicine, National Institutes of Health) database, we found 22 previously published cases of vaginal MA. We thoroughly reviewed the previous literature and summarized the clinical characteristics of 23 patients with vaginal MA in Table 2. The patients’ age ranged from 7 months to 63 years. The most common symptom was vaginal bleeding, reported by nine patients. The greatest dimension of tumor ranged from 0.9–14 cm. Ten of the 11 (90.9%) patients whose stage information was available had International Federation of Gynecology and Obstetrics stage II–III tumors. Fifteen patients underwent surgery, and six received radiation therapy as primary treatment. The follow-up data were available for 17 patients. Three patients developed recurrences; two of them died of multiple metastases, and one was alive with disease. Fourteen patients were alive without evidence of recurrence, with a mean follow-up period of 34.9 (range, 4–103 months) months.

Table 3 summarizes the pathological characteristics, immunophenotypes, and *TP53* mutational profiles of 23 vaginal MA cases. The preoperative biopsy diagnoses were available for 11 patients. The biopsy specimens of five patients were not diagnosed as MA but as moderately or poorly differentiated adenocarcinoma and vaginal adenosis with microglandular hyperplasia or interpreted descriptively as irregular glands with intraluminal invaginations and cystic dilatation. Immunostaining was performed in the six most recent patients. Paired box 2 was positive in the two examined cases, whereas GATA-binding protein 3 (2/3) and cluster of differentiation 10 (4/6) were not expressed in all examined patients. The present case was the only one in which transcription termination factor 1 was evaluated. Five of the six examined patients showed negative estrogen-receptor expression, and progesterone receptor was absent in all six examined patients. Despite the small number of examined patients, the positive rates of mesonephric markers and hormone receptors was similar to that observed in uterine MLA [24]. Information on *TP53* mutation was available only in our case.

We herein present, for the first time, that vaginal MA showed aberrant p53 expression, confirmed with NGS analysis. In particular, we found a truncating *TP53* p.E286* mutation that involves the DNA-binding domain of the p53 protein [25]. This alteration results in a decreased transactivation activity in tumor cells [26], which would then lead to a loss of p53 protein function. We thoroughly searched the PubMed (National Library of Medicine) database to find all previously published cases of malignant mesonephric lesions in which p53 expression and/or *TP53* mutation were examined. Table 4 summarizes the status of p53 expression and *TP53* mutation obtained from 44, 80, and 32 cases of cervical MA, uterine MLA, and ovarian MLA, respectively. Among 11 cases of cervical MA tested with immunostaining, nine were described as having ‘wild-type’ p53 expression pattern, one as ‘totally negative’, and the other as ‘positive’ expression. We were not able to determine whether the latter two cases exhibited wild-type or mutation pattern of p53 immunoreactivity because molecular testing was not performed in either case, and photomicrographs of immunostaining was not shown. Among 38 cervical MA cases with results of genomic profiling, two were reported to harbor *TP53* mutations. The type of *TP53* mutation was p.R280G in one case but not clarified in the other. Eighty cases of uterine MLA were available for p53 expression and *TP53* mutational status. Sixty-nine of the 70 cases tested with p53 immunostaining showed wild-type pattern, while one case was interpreted as ‘negative’; it is unclear whether it referred to a wild-type or mutation pattern. Only one of the 66 cases with molecular profiling results had a pathogenic *TP53* mutation (p.I254N). Twenty-two cases of ovarian MLA showed wild-type p53 expression pattern, and 29 cases did not harbor any pathogenic *TP53* mutation. In summary, *TP53* mutations were identified in three MAs (3/67, 4.5%) and one MLA (1/112, 0.9%). We found that four (2.2%) of the 179 malignant mesonephric lesions of the female genital tract harbored *TP53* mutations.

The prognostic significance of p53 expression in vaginal carcinoma has been investigated [27,28,29,30,31,32], but most did not find any. Moreover, since the reviewed cases predominantly consisted of squamous cell carcinoma rather than adenocarcinoma, the effect of *TP53* mutation on the biological behavior and patient outcomes of vaginal adenocarcinoma remains unknown. In endometrial carcinoma, *TP53*-mutant cases displayed more frequent disease progression and higher mortality than those without *TP53* mutation [33,34]. Human papillomavirus (HPV)-independent endocervical adenocarcinoma harboring *TP53* mutation exhibited deeper invasion, advanced stage, frequent distant metastasis, and shorter survival than HPV-associated, *TP53* wild-type endocervical adenocarcinoma [35]. Furthermore, *TP53* mutations are potential prognostic markers in a variety of malignancies, including pulmonary and pancreatic adenocarcinomas, hepatocellular carcinoma, and renal cell carcinoma [36]. Taken together, it is reasonable to hypothesize that vaginal MA harboring *TP53* mutation may behave more aggressively than *TP53* wild-type MA. In the present case, we could not analyze the long-term outcome of our patient due to the short postoperative period. We anticipate that the clinical and prognostic implications of *TP53* mutation in MA could be investigated in the near future.

In the present case, we observed several areas of severe nuclear enlargement and pleomorphism with marked membrane irregularity, all of which is not typical for MA since it typically displays relatively small, uniform nuclei with minimal-to-mild pleomorphism. Recently, we documented a case of dedifferentiated uterine MLA showing missense *TP53* mutation and p53 overexpression [37]. Therein, the undifferentiated carcinoma component showed a uniform and strong nuclear p53 immunoreactivity, while the MLA component exhibited wild-type pattern. Interestingly, the NGS analysis revealed *TP53* mutation in undifferentiated carcinoma but not in MLA. Although we could not find any prognostic significance of *TP53* mutation, we suspect that *TP53* mutation may be associated, in part, with this high-grade nuclear atypia.

Due to the rarity and the lack of randomized prospective data, the optimal treatment approach for primary vaginal carcinoma is unclear [38]. Many of the recommended approaches are extrapolated from the nearby malignancies, such as cervical and vulvar carcinomas. Surgical resection is typically the treatment of choice for early-stage vaginal carcinomas with favorable anatomical locations and size [39]. In case of stage II disease, not only surgical excision but also radiation therapy is considered adequate. Postoperative radiation therapy should be considered in cases of resected locally advanced-stage, high-risk tumors [40]. From a radiation standpoint, intracavitary or interstitial brachytherapy is utilized in addition to external beam radiation therapy because brachytherapy has been shown to result in superior overall survival [41,42]. The use of radio-sensitizing chemotherapy has also been shown to provide superior local control and disease-free survival, particularly for advanced-stage tumors [38]. In the present case, based on the presence of some risk factors, including rare histological type, subvaginal soft tissue extension, and substantial perineural invasion, the patient received postoperative adjuvant cisplatin-based chemotherapy followed by whole-pelvic radiation therapy and intracavitary brachytherapy. Since there has been no available database for comparing the treatment response to chemoradiation therapy between MA and other histological types of vaginal carcinoma, we cannot conclude that any treatment option is more effective for vaginal MA than the other. Further investigation with a larger cohort is necessary.

## 4. Conclusions

We described the first case of vaginal MA showing nonsense *TP53* mutation and complete lack of p53 immunoreactivity. The diagnosis of vaginal MA was confirmed by the characteristic histological features, immunostaining results, and *KRAS* mutation. *TP53* mutation and aberrant p53 expression were extremely rare findings in MA; according to our review of the previous literature, it accounts for 2.2% of malignant mesonephric lesions arising in the female genital tract. Our observations of unusual immunophenotype and mutational profile in vaginal MA will enrich the knowledge of this rare but important entity, allowing pathologists to make a correct diagnosis and helping clinicians to improve patient outcomes.

## Figures and Tables

**Figure 1 diagnostics-12-00119-f001:**
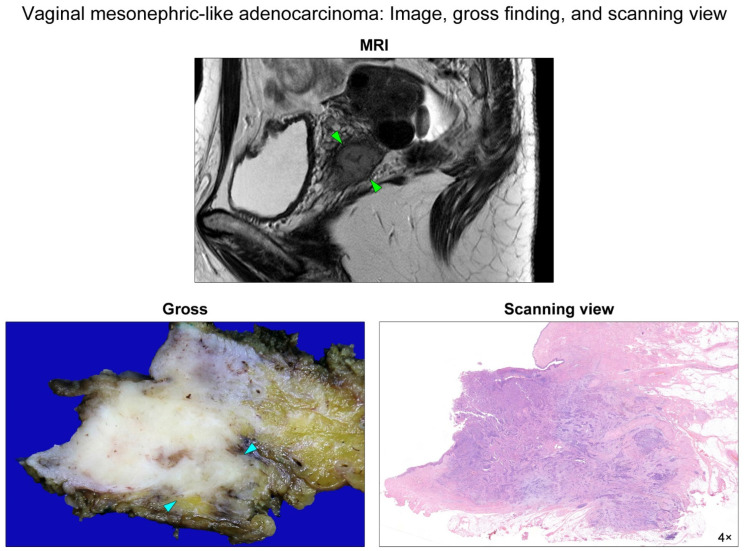
Pelvic magnetic resonance imaging and gross finding and scanning-view photomicrograph. A sagittal T2-weighted image revealed a 2.5-cm mass arising from the left upper vagina (green arrowheads). The mass extended into the extravaginal tissue. The cut section of the resected specimen displayed a poorly circumscribed, yellow-to-white, solid, rubbery tumor involving the subvaginal soft tissue (blue arrowheads). A photomicrograph matched the gross image showing an infiltrative tumor tissue destructively invading through the entire vaginal wall into the subvaginal soft tissue.

**Figure 2 diagnostics-12-00119-f002:**
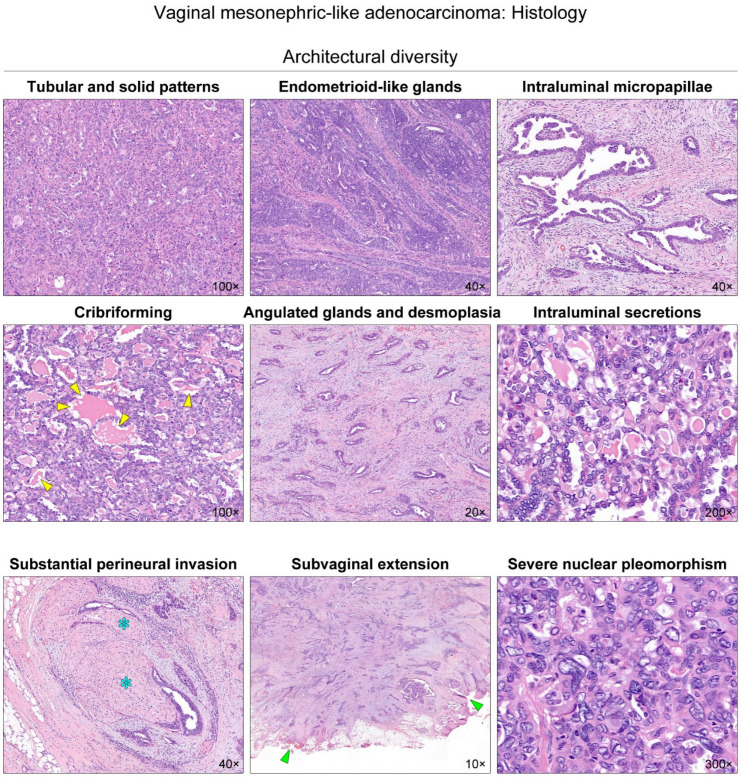
Histological features of vaginal mesonephric adenocarcinoma (MA). The tumor exhibited diverse growth patterns, including compact aggregation or fusion of small tubules, solid cellular sheets, endometrioid-like glands and ducts, papillary and micropapillary architecture, cribriform structure, and scattered, angulated glands associated with prominent desmoplastic stroma. The tubules and glands possessed hyaline-like, densely eosinophilic intraluminal secretions. These secretions had a sharp luminal contour or showed occasional vacuoles that resembled peripheral scalloping of colloid observed in thyroid follicles (yellow arrowheads). We also noted some histological features suggesting aggressive behavior. The variable-sized neoplastic glands grew around and within the nerve fibers (blue asterisks). The infiltrating tumor tissue involved the subvaginal soft tissue resection margin (green arrowheads). In addition, areas showing severe nuclear pleomorphism, enlargement, and marked irregularity of nuclear membrane were frequently observed. Since MA typically displays relatively small, uniform nuclei with minimal-to-mild pleomorphism, our observation of high-grade nuclear atypia seems unusual for vaginal MA.

**Figure 3 diagnostics-12-00119-f003:**
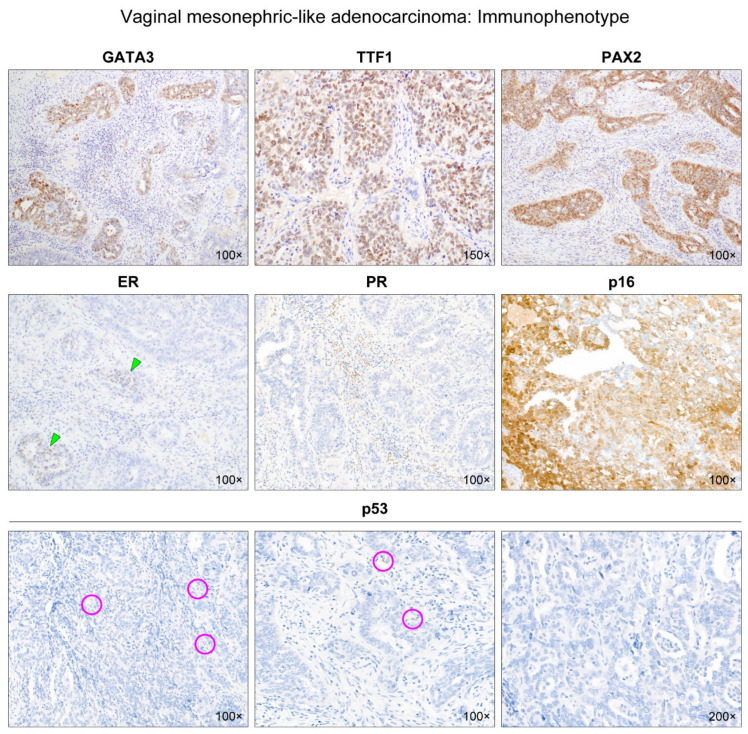
Immunophenotype of vaginal mesonephric adenocarcinoma. We conducted immunostaining for mesonephric markers (GATA-binding protein 3 [GATA3], transcription termination factor 1 [TTF1], paired box 2 [PAX2]), and hormone receptors (estrogen receptor [ER] and progesterone receptor [PR]) p16 and p53. The tumor expressed three mesonephric markers: GATA3, TTF1, and PAX2. The tumor cells were focally positive for GATA3 with moderate staining intensity. TTF1 and PAX2 were diffusely expressed in most of the tumor cells with moderate-to-strong staining intensity. In contrast, ER immunoreactivity was very weak in a few neoplastic glands (green arrowheads), and PR expression was completely absent. p16 positivity was patchy with variable staining intensity. All of the tumor cells exhibited a complete lack of p53 immunoreactivity (mutant p53 expression pattern). A few scattered inflammatory cells and stromal cells showing weak nuclear p53 expression served as positive internal controls (purple circles).

**Table 1 diagnostics-12-00119-t001:** Targeted sequencing results.

Gene	Mutation Type	Sequence Change	Amino Acid Change	Variant Allele Frequency	Clinical Significance
*KRAS*	Missense	c.35G > A	p.G12D	28%	Pathogenic
*TP53*	Nonsense	c.856G > T	p.E286*	39%	Pathogenic

**Table 2 diagnostics-12-00119-t002:** Clinical characteristics of vaginal mesonephric adenocarcinoma.

Case No	Authors (Year Published)	Age	Presenting Symptom or Sign	Tumor Size	Treatment	Initial Stage	Postsurgical Treatment	Recurrence	Follow-Up Period
1	Novak et al. (1954) [10]	42 years	NA	NA	Surgery	NA	NA	NA	NA
2	Novak et al. (1954) [10]	13 years	NA	NA	NA	NA	NA	NA	Died shortly after
3	Novak et al. (1954) [10]	51 years	NA	NA	Surgery (incomplete excision)	NA	NA	NA	NA
4	Novak et al. (1954) [10]	21 years	NA	NA	NA	NA	NA	NA	NA
5	Studdiford (1957) [11]	40 years	NA	NA	Radiation	NA	NA	No	NED (24 months)
6	Studdiford (1957) [11]	16 years	NA	NA	Surgery (incomplete excision)	NA	Radiation	No	NED (60 months)
7	Studdiford (1957) [11]	16 years	NA	NA	Radiation	NA	NA	Yes	Died 2 years later from widespread metastases
8	Studdiford (1957) [11]	42 years	NA	NA	Radiation	NA	NA	No	NED (24 months)
9	Harris and Daly (1966) [12]	61 years	Vaginal bleeding	4 cm	Radiation	NA	NA	No	NED (12 months)
10	Droegemueller et al. (1970) [13]	7.5 years	Vaginal spotting	NA	Surgery (en bloc mass excision)	NA	No	No	NED (53 months)
11	Droegemueller et al. (1970) [13]	8 years	Vaginal bleeding	6 cm	Radiation	III	NA	Yes	Deteriorated (metastases to the pelvis, abdomen, inguinal LN, and lungs)
12	Shaaban (1970) [14]	26 years	Contact bleeding oncoitus or douching	5 cm	Surgery (RH with vaginectomy)	III	No	No	NED (24 months)
13	Siegel et al. (1970) [15]	7 months	Vaginal bleeding	NA	Radiation	III	NA	No	NED (24 months)
14	Hinchey et al. (1983) [16]	29 years	Pelvic fullness	6 cm	Surgery (mass excision with BSO)	NA	Radiation	No	NED (4 months)
15	Bague et al. (2004) [17]	54 years	Enlarged uterus due to leiomyoma	4 cm	Surgery (TH with vaginectomy and BSO)	II	No	Yes	AWD (103 months)
16	Bague et al. (2004) [17]	38 years	Painful coitus	NA	Surgery (mass excision)	NA	NA	NA	NA
17	McNall et al. (2004) [18]	13 years	Vaginal bleeding	6 cm	Surgery (mass excision with partial Vaginectomy, BO, right iliac LNS, and left iliac LND)	III	CCRT	No	NED (55 months)
18	Ersahin et al. (2005) [19]	55 years	Asymptomatic vaginal mass	0.9 cm	Surgery (radical upper vaginectomy with BSO, pelvic LND, and para-aortic LNS)	III	CCRT	No	NED (36 months)
19	Bifulco et al. (2008) [20]	58 years	Pelvic pain and vulvar pruritus	14 cm	Surgery (radical mass excision with pelvic and para-aortic LND)	III	No	No	NED (12 months)
20	Roma (2014) [21]	58 years	Vaginal bleeding	5 cm	Surgery (pelvic exenteration)	III	NA	NA	NA
21	Mueller et al. (2016) [22]	54 years	Vaginal bleeding	2.5 cm	Surgery (mass excision)	II	CCRT	No	NED (48 months)
22	Shoeir et al. (2018) [23]	63 years	Painless vaginal swelling	3.1 cm	Surgery (mass excision)	I	Radiation	No	NED
23	Lee et al. (2021) (the present case)	52 years	Vaginal bleeding	2 cm	Surgery (radical resection with BSO and PLND)	II	SCRT	No	NED (10 months)

Abbreviations: AWD, alive with disease; BO, bilateral oophoropexy; BSO, bilateral salpingo-oophorectomy; LND, lymph node dissection; LNS, lymph node sampling; NA, not applicable; NED, no evidence of disease; RH, radical hysterectomy; SCRT, sequential chemoradiation therapy; TH, total hysterectomy.

**Table 3 diagnostics-12-00119-t003:** Pathological characteristics, immunophenotypes, and *TP53* mutational status of vaginal mesonephric adenocarcinoma.

Case No	Authors (Year Published)	Preoperative Biopsy Diagnosis	Final Diagnosis	PAX8	PAX2	GATA3	TTF1	CD10	ER	PR	PTEN	p16	p53	*TP53* Mutation
1–4	Novak et al. (1954) [10]	NA	NA	NA	NA	NA	NA	NA	NA	NA	NA	NA	NA	NA
5–8	Studdiford (1957) [11]	NA	NA	NA	NA	NA	NA	NA	NA	NA	NA	NA	NA	NA
9	Harris and Daly (1966) [12]	MA	MA	NA	NA	NA	NA	NA	NA	NA	NA	NA	NA	NA
10	Droegemueller et al. (1970) [13]	Papillary MA	Papillary MA	NA	NA	NA	NA	NA	NA	NA	NA	NA	NA	NA
11	Droegemueller et al. (1970) [13]	Clear cell MA	NA	NA	NA	NA	NA	NA	NA	NA	NA	NA	NA	NA
12	Shaaban (1970) [14]	Variable-sized, irregular glands with intraluminal invaginations and marked cystic dilatation	MA	NA	NA	NA	NA	NA	NA	NA	NA	NA	NA	NA
13	Siegel et al. (1970) [15]	MA	NA	NA	NA	NA	NA	NA	NA	NA	NA	NA	NA	NA
14	Hinchey et al. (1983) [16]	NA	NA	NA	NA	NA	NA	NA	NA	NA	NA	NA	NA	NA
15	Bague et al. (2004) [17]	NA	MA	NA	NA	NA	NA	NA	NA	NA	NA	NA	NA	NA
16	Bague et al. (2004) [17]	NA	MCS	NA	NA	NA	NA	NA	NA	NA	NA	NA	NA	NA
17	McNall et al. (2004) [18]	MA	MA	NA	NA	NA	NA	NA	NA	NA	NA	NA	NA	NA
18	Ersahin et al. (2005) [19]	Infiltrating adenocarcinoma	MA	NA	NA	NA	NA	Negative	Negative	Negative	NA	NA	NA	NA
19	Bifulco et al. (2008) [20]	Moderately differentiated adenocarcinoma	MA	NA	NA	NA	NA	Positive	Negative	Negative	NA	NA	NA	NA
20	Roma (2014) [21]	MCS	MCS	Positive	Positive	Focal positive	NA	Focal positive	Negative	NA	NA	Negative	NA	NA
21	Mueller et al. (2016) [22]	Vaginal adenosis with microglandular hyperplasia	MA	NA	NA	NA	NA	Weak positive	Negative	Negative	NA	NA	NA	NA
22	Shoeir et al. (2018) [23]	NA	MA	Diffuse positive	NA	Negative	NA	Focal positive	Negative	Negative	NA	Focal positive	NA	NA
23	Lee et al. (2021) (the present case)	Poorly differentiated adenocarcinoma	MA	Diffuse strong positive	Diffuse strong positive	Focal moderate positive	Diffuse strong positive	Negative	Focal Weak positive	Negative	Noloss	Patchy positive	Mutation pattern (complete absence)	p.E286*

Abbreviations: CD10, cluster of differentiation 10; ER, estrogen receptor; GATA3, GATA-binding protein 3; MA, mesonephric adenocarcinoma; MCS, mesonephric carcinosarcoma; NA, not applicable; PAX2, paired box 2; PAX8, paired box 8; PR; progesterone receptor; PTEN, phosphatase and tensin homolog deleted on chromosome 10; *TP53*, tumor protein 53; TTF1, transcription termination factor 1.

**Table 4 diagnostics-12-00119-t004:** p53 expression and *TP53* mutation of cervical mesonephric and uterine/ovarian mesonephric-like adenocarcinomas.

Organ	Authors (Year Published)	p53 Expression (Number of Examined Cases)	*TP53* Mutation (Number of Mutant/Examined Cases)	Type of *TP53* Mutation
Cervix	Fukunaga et al. (2008) [43]	Positive (1) *	NA	
	Roma (2014) [21]	Wild-type pattern (1)	NA	
	Mirkovic et al. (2015) [44]	NA	1/13	NA
	Kir et al. (2016) [45]	Totally negative (1) *	NA	
	Cavalcanti et al. (2017) [46]	NA	0/1	
	Montalvo et al. (2019) [47]	Wild-type pattern (1)	0/1	
	Skala et al. (2020) [48]	NA	0/1	
	Kim et al. (2020) [4]	NA	0/4	
	Lin et al. (2020) [49]	NA	1/10	p.R280G
	Marani et al. (2021) [50]	Wild-type pattern (1)	0/1	
	Xie et al. (2021) [51]	Wild-type pattern (2)	NA	
	da Silva et al. (2021) [52]	Wild-type pattern (4)	0/8	
Uterine corpus	Ordi et al. (2001) [53]	Wild-type pattern (1)	NA	
	Montagut et al. (2003) [54]	Wild-type pattern (1)	NA	
	Wani et al. (2008) [55]	Wild-type pattern (weak and focal positivity) (1)	NA	
	Mirkovic et al. (2015) [44]	NA	0/2	
	Zhao et al. (2016) [56]	Negative (1) *	NA	
	Kim et al. (2016) [57]	Wild-type pattern (1)	NA	
	McFarland et al. (2016) [6]; Mirkovic et al. (2018) [58]	Wild-type pattern (4)	0/3	
	Ando et al. (2017) [59]	Wild-type pattern (1)	NA	
	Patel et al. (2019) [60]	NA	0/1	
	Yano et al. (2019) [61]	Wild-type pattern (1)	0/1	
	Zhang et al. (2019) [62]	Wild-type pattern (1)	NA	
	Na et al. (2019) [5]	Wild-type pattern (11)	0/11	
	Kolin et al. (2019) [63]	Wild-type pattern (3)	0/4	
	Liang et al. (2020) [64]	Wild-type pattern (2)	0/2	
	Horn et al. (2020) [65]	Wild-type pattern (4)	0/4	
	Xie et al. (2021) [51]	Wild-type pattern (5)	NA	
	da Silva et al. (2021) [52]	Wild-type pattern (6)	1/13	p.I254N
	Choi et al. (2021) [37]	Wild-type pattern (1)	0/1	
	Kim et al. (2021) [24]	Wild-type pattern (25)	0/25	
Ovary	Mirkovic et al. (2015) [44]	NA	0/1	
	McFarland et al. (2016) [6]; Mirkovic et al. (2018) [58]	Wild-type pattern (3)	0/4	
	Chapel et al. (2018) [66]	Wild-type pattern (1)	0/1	
	Seay et al. (2020) [67]	Wild-type pattern (1)	0/1	
	Dundr et al. (2020) [68]	Wild-type pattern (1)	0/1	
	McCluggage et al. (2020) [69]	Wild-type pattern (1)	0/1	
	Chen et al. (2020) [70]	Wild-type pattern (1)	NA	
	Xie et al. (2021) [51]	Wild-type pattern (2)	NA	
	Deolet et al. (2021) [71]	Wild-type pattern (4)	0/5	
	da Silva et al. (2021) [52]	Wild-type pattern (7)	0/15	

* We could not determine whether the p53 expression is either wild-type or mutation pattern. Abbreviations: NA, not applicable; *TP53*, tumor protein 53.

## Data Availability

Not applicable.

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
