# Peer review of "Mesonephric Adenocarcinoma of the Vagina Harboring TP53 Mutation"

_diagnostics, 2022, doi:10.3390/diagnostics12010119_

Round 1
Reviewer 1 Report
The authors described a case of mesonephric adenocarcinoma arising in the vagina with an aberrant expression / mutation of the TP53 protein. Their case report is of interest, but I have some comments which should be considered and addressed by the authors first.
Major comments
1) The authors performed targeted sequencing of the KRAS and TP53 genes. However, as the authors emphasized, their case is the 23rd described in the literature and despite being rare, it does not represent a novel observation. The impact of their observation could be increased by complex genomic profiling of the tumor. After that, it would be of interest to compare their data with literary data of tumors arising in the cervix, uterine corpus, and ovary.
2) In the Title, the authors emphasized that this is “The First Description of Pathogenic TP53 Mutation and Aberrant p53 Expression”. Although this statement is true, I am not convinced that the presence of TP53 mutation in MA deserved such emphasis. The authors stated (Introduction, line 68-69) that they observed "mutation and aberrant p53 expression, both of which have never been reported in previous literature on vaginal MA." This is again true, but one of the possible explanations for this is that p53 testing has simply not been performed in the previous vaginal MA cases. In other locations, the TP53 mutation occurs in about 5-10% of MA cases (see for example: Lin DI, et al. Gynecol Oncol Rep. 2020;34:100652), so the authors finding it in a vaginal MA is not so surprising. However, according to their interpretation this seems to be the main finding as is emphasized at the end of the Discussion and Conclusion sections.
3) In the Conclusion section (line 190) the authors the authors stated that "TP53 mutation and aberrant p53 expression were extremely rare findings in MA." I disagree with this statement, TP53 mutation has been described in up to 10% of MA cases (see the comment and reference in point 1). The authors should modify their Discussion / Conclusion sections and support their statements by relevant references.
Minor points
1) Abstract, line 15, and Case presentation, line 80: the authors stated that “Based on the result of the punch biopsy performed, poorly differentiated adenocarcinoma was observed”. Do the authors mean diagnosed?
2) Introduction: the authors stated that MA can mimic more common gynaecological malignancies and wrong diagnosis can lead to incorrect treatment. Could the authors briefly mention, from their point of view, how is the treatment of MA different from other carcinomas arising in this location?
Author Response
Major comment 1) The authors performed targeted sequencing of the KRAS and TP53 genes. However, as the authors emphasized, their case is the 23rd described in the literature and despite being rare, it does not represent a novel observation. The impact of their observation could be increased by complex genomic profiling of the tumor. After that, it would be of interest to compare their data with literary data of tumors arising in the cervix, uterine corpus, and ovary.
Answer 1) Thank you for your valuable suggestion. We confirmed the presence of both KRAS and TP53 mutations by next-generation sequencing technique using the Oncomine Comprehensive Assay v3 (Thermo Fisher Scientific), a commercial, amplicon-based, targeted cancer panel that enables the detection of relevant single-nucleotide variants, amplifications, and indels from 161 unique genes. We found a truncating (nonsense) TP53 p.E286* mutation with a variant allele frequency of 39%. To compare our findings with previous data, we have searched 179 malignant mesonephric lesions from the PubMed database and gathered the detailed information on the status of p53 expression and TP53 mutation. We showed that TP53 mutations were identified in three cervical/vaginal MAs (3/67, 4.5%) and one uterine/ovarian MLA (1/112, 0.9%). In total, four (2.2%) of the 179 malignant mesonephric lesions of the female genital tract harbored TP53 mutations. We considered TP53 mutation a very uncommon phenomenon in malignant mesonephric lesions. We created new table (Table 4) and paragraphs in the Discussion section.
Major comment 2) In the Title, the authors emphasized that this is “The First Description of Pathogenic TP53 Mutation and Aberrant p53 Expression”. Although this statement is true, I am not convinced that the presence of TP53 mutation in MA deserved such emphasis. The authors stated (Introduction, line 68-69) that they observed "mutation and aberrant p53 expression, both of which have never been reported in previous literature on vaginal MA." This is again true, but one of the possible explanations for this is that p53 testing has simply not been performed in the previous vaginal MA cases. In other locations, the TP53 mutation occurs in about 5-10% of MA cases (see for example: Lin DI, et al. Gynecol Oncol Rep. 2020;34:100652), so the authors finding it in a vaginal MA is not so surprising. However, according to their interpretation this seems to be the main finding as is emphasized at the end of the Discussion and Conclusion sections.
Answer 2) We appreciate your comment. We reviewed the article that you kindly mentioned, as well as others reporting p53 immunostaining results and/or TP53 mutational status. As mentioned above, we found a total of 179 cases of malignant mesonephric lesions from the PubMed database. We also deepened our discussion by referring to other literatures that studied the association of p53 expression status and prognosis of patients with some different types of gynecological malignancy. We changed the nuance of the Discussion and Conclusion sections to lay out our findings more appropriately.
Major comment 3) In the Conclusion section (line 190) the authors the authors stated that "TP53 mutation and aberrant p53 expression were extremely rare findings in MA." I disagree with this statement, TP53 mutation has been described in up to 10% of MA cases (see the comment and reference in point 1). The authors should modify their Discussion / Conclusion sections and support their statements by relevant references.
Answer 3) Thank you for your comment. From our extensive review of the previous literature, we concluded that only 2.2% (4/179) of malignant mesonephric lesions harbor pathogenic TP53 mutations. As mentioned above, we rewrote the Discussion section and modified the Conclusion section.
Minor comment 4) Abstract, line 15, and Case presentation, line 80: the authors stated that “Based on the result of the punch biopsy performed, poorly differentiated adenocarcinoma was observed”. Do the authors mean diagnosed?
Answer 4) Thank you for your comment. We did mean ‘diagnosed’, and corrected the sentence.
Minor comment 5) Introduction: the authors stated that MA can mimic more common gynaecological malignancies and wrong diagnosis can lead to incorrect treatment. Could the authors briefly mention, from their point of view, how is the treatment of MA different from other carcinomas arising in this location?
Answer 5) Thank you for your comment. Both MA and MLA are often misdiagnosed as other histological types of adenocarcinoma arising in the female genital tract including uterine cervix, corpus, and ovary as well as the vagina. Nevertheless, the treatment options for vaginal MA are not significantly different from those of other histological types of vaginal carcinoma due to its rarity and the lack of randomized prospective data. In the present case, based on the presence of some risk factors including rare histological type, subvaginal soft tissue extension, and substantial perineural invasion, the patient received postoperative adjuvant cisplatin-based chemotherapy followed by whole-pelvic radiation therapy and intracavitary brachytherapy. We believe that sequential chemoradiation therapy might have contributed to the current patient’s disease-free status. We explained the current treatment options of vaginal carcinoma and our limitations in the last paragraph of Discussion section.
Reviewer 2 Report
The authors described significantly rare case of Mesonephric adenocarcinoma with vaginal location. In fact, as indicated in the title, it is the first time when such pathology is described together with presentation of aberrant p53 expression in tumor tissue. Thus, the paper presented undoubtedly demonstrate novelty.
A few comments that appeared when revising the paper:
- I think that the title could be limited to the key message, and the fragment on brief review of the current literature could be omitted there. Of course, the revision part included within paper should remain within the paper as beneficial support for the discussion section.
- If available, could the authors describe more precisely therapeutic approaches implemented in the case report subject following diagnosis? In addition, short information on patientscondition including steps performed in the course of post-treatment monitoring would be essential.
- Regarding the results presented in figure 1 to 3, I would recommend including scale on images or description of the microscope setting within the legend where applicable.
- What eventual limitations could the authors hihglight regarding presented case data or the currently available data, that could be emphasized within ‘Discussion’ section?
Author Response
Comment 1) I think that the title could be limited to the key message, and the fragment on brief review of the current literature could be omitted there. Of course, the revision part included within paper should remain within the paper as beneficial support for the discussion section.
Answer 1) Thank you for your suggestion. We changed our title and made it shorter. We also created a new table (Table 4) and rewrote the Discussion section to lay out our findings more appropriately.
Comment 2) If available, could the authors describe more precisely therapeutic approaches implemented in the case report subject following diagnosis? In addition, short information on patient’s condition including steps performed in the course of post-treatment monitoring would be essential.
Answer 2) Thank you for your suggestion. We reviewed the electronic medical records for detailed treatment regimen and also consulted our gynecologists to inquire the patient’s condition. We added the detailed information on the postoperative course in the last paragraph of Case Presentation section.
Comment 3) Regarding the results presented in figure 1 to 3, I would recommend including scale on images or description of the microscope setting within the legend where applicable.
Answer 3) Thank you for your recommendation. We added the original magnification in the right lower corner of each photomicrograph image.
Comment 4) What eventual limitations could the authors highlight regarding presented case data or the currently available data, that could be emphasized within ‘Discussion’ section?
Answer 4) The eventual limitation is that we could not find a prognostic significance of TP53 mutation in our vaginal MA case due to the short follow-up period. We could not suggest the appropriate management strategies for advanced-stage vaginal MA due to its rarity and the lack of randomized prospective data. The Discussion section has been largely modified by in-depth review of previous literature. A new paragraph was added to explain the importance and potential prognostic significance of TP53 mutation.
Round 2
Reviewer 1 Report
Thank you for addressing all my points. I have no other comments.